# Bacterial DNAemia in Alzheimer’s Disease and Mild Cognitive Impairment: Association with Cognitive Decline, Plasma BDNF Levels, and Inflammatory Response

**DOI:** 10.3390/ijms24010078

**Published:** 2022-12-21

**Authors:** Robertina Giacconi, Patrizia D’Aquila, Marta Balietti, Cinzia Giuli, Marco Malavolta, Francesco Piacenza, Laura Costarelli, Demetrio Postacchini, Giuseppe Passarino, Dina Bellizzi, Mauro Provinciali

**Affiliations:** 1Advanced Technology Center for Aging Research, IRCCS INRCA, 60121 Ancona, Italy; 2Department of Biology, Ecology and Earth Sciences (DIBEST), University of Calabria, 87036 Rende, Italy; 3Center for Neurobiology of Aging, IRCCS INRCA, 60121 Ancona, Italy; 4Geriatrics Operative Unit, IRCCS INRCA, 63900 Fermo, Italy; 5Clinical Laboratory & Molecular Diagnostics, IRCCS INRCA, 60121 Ancona, Italy

**Keywords:** circulating bacterial DNA, inflammation, BDNF, Alzheimer’s disease, mild cognitive impairment

## Abstract

Microbial dysbiosis (MD) provokes gut barrier alterations and bacterial translocation in the bloodstream. The increased blood bacterial DNA (BB-DNA) may promote peripheral- and neuro-inflammation, contributing to cognitive impairment. MD also influences brain-derived neurotrophic factor (BDNF) production, whose alterations contribute to the etiopathogenesis of Alzheimer’s disease (AD). The purpose of this study is to measure BB-DNA in healthy elderly controls (EC), and in patients with mild cognitive impairment (MCI) and AD to explore the effect on plasma BDNF levels (pBDNF), the inflammatory response, and the association with cognitive decline during a two-year follow-up. Baseline BB-DNA and pBDNF were significantly higher in MCI and AD than in EC. BB-DNA was positively correlated with pBDNF in AD, plasma Tumor necrosis factor-alpha (TNF-α), and Interleukin-10 (IL-10) levels in MCI. AD patients with BB-DNA values above the 50th percentile had lower baseline Mini-Mental State Examination (MMSE). After a two-year follow-up, AD patients with the highest BB-DNA tertile had a worse cognitive decline, while higher BB-DNA levels were associated with higher TNF-α and lower IL-10 in MCI. Our study demonstrates that, in early AD, the higher the BB-DNA levels, the higher the pBDNF levels, suggesting a defensive attempt; BB-DNA seems to play a role in the AD severity/progression; in MCI, higher BB-DNA may trigger an increased inflammatory response.

## 1. Introduction

Alzheimer’s disease (AD) is the most prevalent form of dementia in the elderly; its pathogenesis is not completely understood, and there is no effective cure [1]. Nevertheless, several modifiable risk factors, including hypertension, obesity, physical inactivity, smoking, excessive alcohol intake, and stress have been identified [2].

Recent research has shown that gut dysbiosis, which affects brain immune homeostasis via the microbiota–gut–brain axis, may play a significant role in the pathogenesis of neurodegenerative diseases [3].

Increased gut permeability (leaky gut) may be the primary source of bacterial DNA in the bloodstream [4], in addition to that from the skin, oral cavity, and reproductive and respiratory tracts [5,6,7,8]. Although the level of blood bacterial DNA (BB-DNA) is not associated with sepsis [5], its increase can induce the onset of several pathologies, including type-2 diabetes, cardiovascular, and chronic kidney diseases [4,7,9,10,11].

Evidence shows that BB-DNA can trigger alterations in the blood–brain barrier (BBB), neurotoxic protein aggregation, hyperactivation of the innate immune system, and neuroinflammation [12,13,14,15,16,17,18,19], thus contributing to cognitive impairment. Studies in mice have demonstrated that a reduction in gut dysbiosis due to the regulation of inflammatory and anti-inflammatory cytokine expression, may delay cognitive deficits [20], whereas a decrease in IL-10 production over time is associated with worse cognitive decline [21]. In addition to affecting the inflammatory balance, gut dysbiosis affects other pathways, including BDNF production, neurotransmitter metabolism, neuronal functioning, and structural integrity [22,23,24].

Despite these findings, little is known about the potential intermediary role of BB-DNA in the etiopathogenesis of age-related neurodegenerative diseases, where neuroinflammation and altered trophic support to neurons have been widely reported [25,26]. As a result, the present study aims to investigate the association between BB-DNA and cognitive decline in patients with mild cognitive impairment (MCI) and AD during a two-year follow-up, as well as the influence of BB-DNA on peripheral inflammatory response and plasma BDNF levels, which may provide information on the neuronal neurotrophin [22].

## 2. Results

### 2.1. Baseline Characteristics of EC, MCI and AD Patients

Table 1 summarizes the socio-demographic and neuropsychological profiles, lifestyle, and medication usage of the cohorts enrolled in the study.

The MCI and AD patients were significantly older (*p* < 0.0001) and with a lower percentage of females (*p* < 0.05) than the EC group. AD patients had lower MMSE scores (*p* < 0.001), physical activity (*p* < 0.0001), social networks (*p* < 0.05), and functional capabilities (IADL, ADL; *p* < 0.0001) than MCI and EC subjects, and also had a lower level of education (*p* < 0.0001) than the EC group. MCI patients had lower MMSE scores, IADL, physical activity, and level of education than EC (*p* < 0.001). The prevalence of chronic obstructive pulmonary disease, hypertension, diabetes, coronary heart disease, chronic kidney disease, inflammatory bowel disease, and peptic ulcers did not differ significantly between groups. AD patients had a significantly higher frequency of previous ictus and peripheral artery disease (*p* < 0.05) but a significantly lower frequency of atrial fibrillation and gastritis (*p* < 0.01) than EC. With regard to drug use, only AD patients took acetylcholinesterase inhibitors (AChEIs). The prescription of antidepressants, benzodiazepines, and anticoagulants differed considerably between the EC, MCI, and AD groups (*p* < 0.05).

### 2.2. BB-DNA and Plasma BDNF Levels

Baseline BB-DNA and plasma BDNF levels were significantly higher in MCI and AD patients than in EC subjects (BB-DNA, *p* < 0.05; BDNF, *p* <0.01) (Figure 1). Data were adjusted for age, sex, education, previous ictus, peripheral artery disease, and drug use. BB-DNA was analyzed in each cohort in relation to smoking habits, but without any significant differences (Appendix A).

We investigated the possible association between BB-DNA and plasma BDNF levels in AD and MCI patients using multivariate linear regression that included the main independent variables that may affect BDNF production (i.e., age, sex, AChEI, benzodiazepines, antidepressants, lipid-lowering medications, plasma BDNF levels, alcohol consumption, smoking habits, and PASE score) [22] (Table 2). In the first model, we compared this association between AD patients with mild [Clinical Dementia Rating (CDR) 1] and moderate (CDR2) dementia stages (Table 2). In the second model, we performed multivariate linear regression with enter and backward stepwise regression approaches for the whole AD group (Appendix A). A positive association between BB-DNA and plasma BDNF levels was found in the whole AD group (β coefficient = 0.239 and *p* < 0.05, Appendix A; β coefficient = 0.231 and *p* < 0.05, Appendix A) and in AD patients with CDR2 (β coefficient = 0.407 and *p* < 0.05) (Table 2). There was no significant association between BB-DNA and plasma BDNF levels in AD patients with CDR1 (Table 2) and in MCI subjects (Appendix A).

### 2.3. Cognitive Function in Relation to BB-DNA Percentiles

The evaluation of cognitive function in EC, MCI, and AD subjects based on BB-DNA percentiles revealed that the MMSE scores were comparable between BB-DNA_low_ (values below the 50th percentile) and BB-DNA_high_ (values above the 50th percentile) in EC and MCI individuals, while BB-DNA_high_ AD patients showed a significantly lower MMSE score (Figure 2). Similarly, the ADAS-Cog score tended to increase in BB-DNA_high_ AD patients (*p* = 0.069) (Appendix A). The relationship between BB-DNA percentiles and the attentive matrices test, immediate prose recall test, delayed prose recall test, total prose recall test, word pairing test, semantic verbal fluency test, phonemic verbal fluency test, Corsi supraspan test, forward digit span test, reverse digit span test, and lubben social network scale was also assessed in MCI and AD patients (Appendix A). Except for the reverse digit span test, which was significantly lower in BB-DNA_high_ than in BB-DNA_low_ in both MCI and AD patients, no significant differences were detected.

### 2.4. Plasma Cytokine Levels

Plasma TNF-α and IL-10 levels were measured at baseline and after a two-year follow-up in a subgroup of randomly selected patients. They were subdivided as follows: 55 EC (mean age 73 ± 6 years; 11 males and 44 females), 55 MCI (mean age 76 ± 6 years; 17 males and 38 females), and 65 AD (mean age 76 ± 5 years; 22 males and 43 females).

There was no significant difference in cytokine levels between the cohorts (Appendix A), either at baseline or after 2 years of follow-up. Only EC males exhibited significantly higher IL-10 values at baseline compared to MCI and AD males (*p* < 0.05, Appendix A).

### 2.5. Correlations between BB-DNA and Plasma Cytokines Levels at Baseline and after Two-Year Follow-Up Period

MCI patients had significant positive correlations between BB-DNA and plasma TNF-α levels at baseline (r = 0.270; *p* < 0.05) and after 2 years of follow-up (r = 0.277; *p* < 0.05), as well as between BB-DNA and plasma IL-10 levels at baseline (r = 0.299; *p* < 0.05) (Appendix A). No significant correlations were found in EC and AD subjects.

### 2.6. Change in AD Patients’ ADAS-Cog Score at Two-Year Follow-Up Based on BB-DNA Tertiles

The percent change in the ADAS-cog score [(ADAS-cog score at follow-up − ADAS-cog score at baseline)/ADAS-cog score at baseline) × 100] was calculated to analyze the potential association with BB-DNA tertiles labelled as low, medium, and high. A significantly worse cognitive decline was observed in AD patients with BB-DNA_high_ as compared to those with BB-DNA_low_ and BB-DNA_medium_ (*p* < 0.001 and *p* < 0.05, respectively) (Figure 3).

### 2.7. Changes in Plasma Cytokine and BDNF Levels at Two-Year Follow-Up in EC, MCI, and AD Patients Based on BB-DNA Tertiles

The percent change in cytokine production [(cytokine levels at follow-up − cytokine levels at baseline)/cytokine levels at baseline) × 100] was measured to evaluate the variation of inflammatory and anti-inflammatory response in relation to BB-DNA. In MCI patients, BB-DNA_high_ had a significantly increased TNF-α production and a significantly decreased IL-10 production over time compared to BB-DNA_low_ and BB-DNA_medium_ (*p* < 0.05 and *p* < 0.01, respectively) (Figure 4). No significant difference was observed in EC and AD subjects. The percent change in plasma BDNF levels [(plasma BDNF levels at follow-up − plasma BDNF levels at baseline)/plasma BDNF levels at baseline) × 100] was also measured to investigate the variation of neurotrophin in relation to BB-DNA (Appendix A). No significant change was observed in the three cohorts, suggesting that the baseline up-regulation found in AD patients was a transient response that weakened until it disappeared as the disease worsened [22].

## 3. Discussion

A recent meta-analysis demonstrated a significant relationship between chronic inflammation in the gastrointestinal tract and the onset of AD in the adult population, suggesting that gut microbiota may play a role in the pathogenesis of neurodegenerative diseases [27]. Moreover, changes in the composition of gut microbiota correlate with the severity of AD and the level of cognitive impairment [28]. Microbial dysbiosis and impaired intestinal barrier integrity are among the leading causes of elevated BB-DNA levels [29], although chronic periodontitis may also contribute significantly. In fact, *Porphyromonas gingivalis* has been detected in the brains of AD patients [30] and linked to the development of AD [31]. Regardless of the source (s) and despite previous investigations demonstrating BB-DNA increments in several age-related diseases [9,10,11], there is still no evidence in neurodegenerative pathologies.

For the first time, this study reports that BB-DNA levels are higher in MCI and AD patients than in EC and that the higher the BB-DNA levels, the more severe the cognitive decline in AD patients. The recent finding that extracellular bacterial DNA promotes Tau misfolding and β-Amyloid (βA) aggregation is consistent with and strengthens our results [19,32]. It is interesting to note that BB-DNA levels were comparable in MCI and AD patients, suggesting a potential involvement in the pre-dementia stage and paving the way for its use as an early diagnostic biomarker. In contrast to our previous findings [7], no significant differences were observed in smokers from each group. This may be due to the smaller sample size analyzed or the different age range compared to the previous study, or the influence of other environmental factors.

Plasma BDNF levels followed the same pattern as BB-DNA: MCI and AD patients had higher values than EC subjects. A positive association between BB-DNA and plasma BDNF levels was also found in AD patients, particularly those with more severe dementia (i.e., patients with CDR2). This result supports previous findings that the composition of the microbiota is associated with plasma BDNF concentrations [33], and that bacterial brain infection stimulates the synthesis of cerebral BDNF [34]. Furthermore, the up-regulation of BDNF production is considered to be a neuroprotective response against harmful stimuli [22].

Inflammation is considered to be a major contributor to cognitive decline in the elderly subjects [35]. In this study, we measured the plasma TNF-α and IL-10 levels at baseline and after a two-year follow-up period. In contrast to other research, no significant differences were found between EC, MCI, and AD subjects [35,36], although baseline IL-10 values were higher in EC males compared to MCI and AD males. Several confounding factors such as anti-inflammatory medications, depressive status, and cardiovascular and metabolic conditions [37] distributed variably among cohorts may account for the discrepancy among studies. In this study, however, a higher BB-DNA concentration was associated with a higher TNF-α and a lower IL-10 production over time in MCI patients. Some studies failed to detect a correlation between circulating inflammatory markers and the pathogenesis and progression of AD [38,39]. Nevertheless, recent research has demonstrated that increased IL-10 production delays the transition from MCI to dementia [40], whereas decreasing IL-10 production over time is associated with worse cognitive decline in MCI patients [21]. In addition, TNF-α can modulate BBB permeability and promote Aβ peptide accumulation [41], and elevated plasma levels of this pro-inflammatory cytokine are associated with more severe cognitive decline [42]. These findings may suggest that MCI patients with higher BB-DNA levels, a diminished anti-inflammatory response, and an enhanced inflammatory state are at a greater risk of developing dementia. This study did not examine other pro- and anti-inflammatory mediators that may play a relevant role in intestinal dysbiosis and neurodegenerative diseases. For instance, during chronic microbial dysbiosis certain bacterial species and their metabolites may trigger neuroinflammatory pathways promoting Aβ accumulation [43,44]. The interaction between various species of Aβ with receptors in glial cells induces the release of pro-inflammatory cytokines, leading to BBB dysfunction through increasing permeability, inducing structural changes in brain capillaries, and enhancing the migration of immune cells [45]. Moreover, other anti-inflammatory cytokines may play a protective role in dementia, such as IL-4, whose high circulating levels are associated with better cognitive function [46]. However, contrasting results exist in literature on the role of systemic inflammatory mediators as AD risk factors [47] and the topic deserves further investigations.

This study has some limitations. First, the follow-up duration may be insufficient for studying the disease progression [48]. Second, BB-DNA was only measured at baseline, and its origin has not been determined. Third, plasma cytokines were assessed in a subgroup of patients, and the sample size may have been too small; other pro- and anti-inflammatory mediators should also be assayed to strengthen the findings of TNF-α and IL-10.

This study’s strength is that, to our knowledge, it is the first to evaluate the relationship between BB- DNA, plasma BDNF levels, peripheral inflammatory status, and the progression of cognitive decline in MCI and AD patients.

In conclusion, BB-DNA was higher in MCI and AD than in EC subjects and was positively correlated with baseline plasma BDNF levels in AD patients. AD patients with higher BB-DNA levels demonstrated a greater longitudinal cognitive decline, suggesting that BB-DNA may play a role in the deterioration of these individuals’ cognitive status. Finally, in MCI patients, higher BB-DNA levels were associated with an increase in TNF-α and a decrease in IL-10 production over time. Future research should focus on elucidating whether BB-DNA may exert a causative or indirect effect in the pathophysiology of AD due to the host’s response. Additional research on larger cohorts is also required to confirm our findings and clarify whether BB-DNA may serve as a useful prognostic marker of AD progression and/or the conversion of MCI to dementia.

## 4. Materials and Methods

### 4.1. Participants

The My Mind Project study protocol [49] complied with the principles of the Declaration of Helsinki and was approved by the local Ethics Committee (IRCCS INRCA Bioethics Advisory Committee, Ancona, Italy; code SC/12/301). Written informed consent was obtained from all participants or their caregiver. All participants, who were enrolled between June 2012 and October 2014, underwent a complete clinical, physical, neuropsychological, and functional evaluation. The status of EC was defined as the absence of relevant cognitive diseases; MCI was diagnosed by means of an extensive neuropsychological, clinical, and functional assessment, as well as by neuroimaging and laboratory tests in accordance with diagnostic guidelines [50]. DSM-IV or NINCDS-ADRDA criteria were used to diagnose possible or probable AD [51], and the clinical dementia rating scale (CDR) [52] was utilized to evaluate the staging of AD.

The inclusion criteria were the following: age 65 or older, availability during testing phases, the presence of a caregiver for subjects with cognitive decline (specific criterion for MCI and AD patients), and the ability to sign the informed consent (subjects who were not able to speak, were incapacitated or in need of a support administrator). The exclusion criteria were the following: serious medical problems (i.e., recent cardiovascular events, cancer, infections or acute renal failure) or major psychiatric disorders (i.e., major depression, schizophrenia, and other psychiatric illnesses that could limit participation in the study), sensorimotor deficits, severe AD, and the presence of neurodegenerative disorders other than AD (e.g., Parkinson’s disease, progressive supranuclear palsy, cortico-basal degeneration, Lewy’s body dementia, Huntington’s disease, frontotemporal dementia, Pick’s disease) cerebral hypoxia (acute or chronic), infections of the CNS (i.e., brain abscess, meningitis, AIDS), Creutzfeldt-Jacob disease, brain cancers, untreated epilepsy, and alcohol or drug abuse in the past year.

Laboratory parameters, plasma cytokines levels, plasma BDNF amount, and a comprehensive clinical and neuropsychological assessment were carried out at baseline and after two years; BB-DNA was only determined at baseline. A subgroup of subjects was randomly selected for the analysis of plasma cytokines levels as follows: 55 EC (mean age 73 ± 6 years; 11 males and 44 females), 55 MCI (mean age 76 ± 6 years; 17 males and 38 females), and 65 AD (mean age 76 ± 5 years; 22 males and 43 females) patients.

The use of the following drugs was also registered: acetylcholinesterase inhibitors (AChEIs), benzodiazepines, antidepressants, anticoagulants/antiplatelets drugs, antihypertensives, nonsteroidal anti-inflammatory drugs (NSAIDs), lipid lowering medications, oral hypoglycemic medications, diuretics, drugs for thyroid dysfunctions and corticosteroids.

### 4.2. Neuropsychological and Functional Assessment

The neuropsychological test battery was previously described in Giuli et al. [49]. Psychosocial (geriatric depression scale [GDS], lubben social network scale [LSNS], and perceived stress scale [PSS]), functional (activities of daily living [ADL], and instrumental activities of daily living [IADL]), and cognitive function (Mini Mental State Examination [MMSE], clinical dementia rating [CDR], and ADAS-cog) were evaluated. Further cognitive tests were applied (i.e., forward and reverse digit span, Corsi supraspan, attentive matrices, semantic and phonemic verbal fluency, immediate/delayed/total prose recall, and word pairing learning tests). Data about lifestyle (physical activity scale for elderly [PASE], smoking habits, alcohol consumption, and body mass index [BMI]) were also recorded.

### 4.3. Determination of BB-DNA

DNA was extracted from 500 µL of biobank whole blood samples using a QIAamp DNA Blood Mini Kit (Qiagen GmbH, Hilden, Germany) in accordance with the manufacturer’s instructions.

Details concerning the quantification of 16S rRNA have been previously reported [53]. Briefly, highly sensitive and specific universal primers targeting the V3–V4 hypervariable region of the bacterial 16S rDNA were used in real-time qPCR reactions to quantify the 16S rRNA gene copy number in DNA samples. The PCR mixture (20 µL) consisted of 20 ng of DNA, SensiFAST SYBR Hi-ROX Mix 1X (Bioline, London, UK), and 0.4 µM of the following primers: For 5′-TCCTACGGGAGGCAGCAGT-3′ and Rev 5′-GGACTACCAGGGTATCTAATCCTGTT-3′.

The thermal profile used for the reaction included a heat activation of the enzyme at 95 °C for 2 min, followed by 40 cycles of denaturation at 95 °C for 15 s and annealing/extension at 60 °C for 60 s, followed by melt analysis ramping at 60–95 °C. All measurements were taken in the log phase of amplification. Standard curves obtained using a ten-fold dilution series of bacterial DNA standards (Femto bacterial DNA quantification kit, Zymo research, Irvine, CA, USA) ranging from 2 × 10^−4^ to 2 pg were routinely run with each sample set and compared to previous standard curves to check for consistency between runs. The quality of amplicon was ascertained via melting curves. The amplification of samples and standard dilutions were performed in triplicate on the QuantStudio3 Real-Time PCR System (Applied Biosystems by Life Technologies, Carlsbad, CA, USA). BB-DNA levels were expressed as ng/mL of whole blood and were calculated by normalizing the absolute quantities of BB-DNA of each sample to their dilution factors and to the volume of starting blood used for the extraction.

### 4.4. Determination of Plasma BDNF

A part of the data has already been published [54,55]. Briefly, lithium heparin whole blood was drawn between 8:00 and 9:00 a.m. from each subject in a fasting state. Blood was centrifuged at 200× *g* for 10 min, and plasma was separated from platelets by centrifuging the platelet-rich plasma at 1500× *g* for 15 min. Plasma was immediately stored at −80 °C until use. Plasma BDNF was assayed by using a commercial kit (BDNF Human ELISA kit, ab99978; Abcam, Cambridge, UK) in accordance with the manufacturer’s instructions. All samples were tested in duplicate, and results were expressed as ng/mL.

### 4.5. Plasma TNF-α and IL-10 Measurements

Plasma cytokine concentrations were measured using the Human TNF alpha ELISA kit (ab181421; Abcam, Cambridge, UK) and the Human IL-10 ELISA Kit (ab185986; Abcam, Cambridge, UK). ELISA assays were performed in accordance with the manufacturer’s instructions. All samples were tested in duplicate, and results were expressed as pg/mL.

### 4.6. Statistical Analysis

Data were reported as the mean ± standard error of the mean (SEM) or as percentages for continuous and categorical variables, respectively. For continuous variables, normal distribution was verified by the 1-sample Kolmogorov–Smirnov test. All non-normally distributed variables were log-transformed. The differences in BB-DNA between groups were analyzed by ANCOVA analysis and generalized linear models (linear model with log-transformed values and identity link-function) after adjusting for age, sex, education, previous ictus, peripheral artery disease and drug use. Pearson’s χ2 test was applied for categorical variables. Spearman correlation between BB-DNA and plasma cytokine levels was also used. The relationship between BB-DNA and plasma BDNF levels was analyzed using linear regression analysis employing the enter and stepwise methods. The variables considered in the enter method were sex, age, AChEIs, benzodiazepines, antidepressants, lipid-lowering medications, alcohol consumption, smoking habits, and PASE. Linear regression analysis using the stepwise method included the following covariates: sex, age, AChEIs, benzodiazepines, antidepressants, lipid-lowering medications, alcohol consumption, smoking habits, PASE, BMI, education, IADL, GDS, and the ADAS-cog score. Each variable that met the criteria for removal was eliminated in a stepwise manner. Only variables below the removal threshold were retained in the final step. An ANCOVA analysis, adjusted for age, sex, education, and drug use, was employed to evaluate the differences in BB-DNA and plasma BDNF levels among cohorts. An ANCOVA analysis, adjusted for age, sex, education, and IADL, was also employed to analyze the baseline MMSE score, changes in the ADAS-cog score, plasma cytokines, and plasma BDNF levels in relation to BB-DNA percentiles. All analyses were performed using the SPSS/Win program (version 22.0; Spss Inc., Chicago, IL, USA).

## Figures and Tables

**Figure 1 ijms-24-00078-f001:**
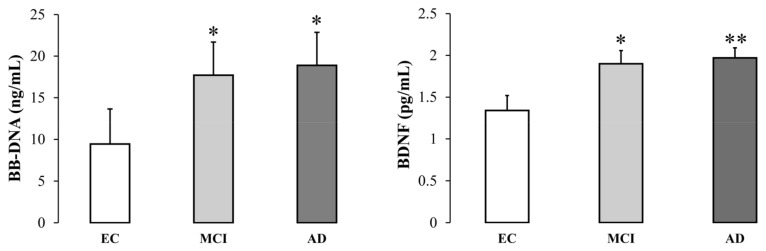
Blood bacterial DNA (BB-DNA) and plasma BDNF levels from elderly controls (EC), and mild cognitive impairment (MCI) and Alzheimer’s disease (AD) subjects. AD and MCI patients had significantly higher baseline BB-DNA and plasma BDNF levels as compared to EC. ANCOVA analysis and generalized linear models (linear model with log-transformed values and identity link-function) correcting for age, sex, education, previous ictus and peripheral artery disease and drug use were applied. * *p* < 0.05, ** *p* < 0.01 as compared to EC.

**Figure 2 ijms-24-00078-f002:**
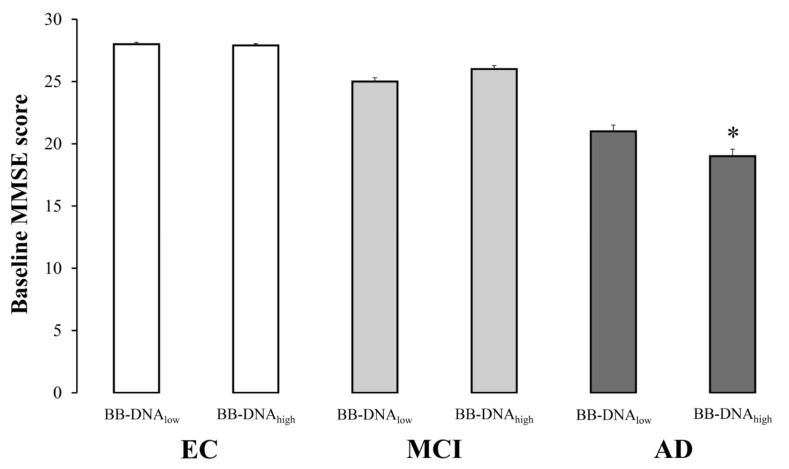
The baseline Mini Mental State Examination (MMSE) scores in elderly controls (EC), mild cognitive impairment (MCI) and Alzheimer’s disease (AD) subjects in accordance with blood bacterial DNA (BB-DNA) percentiles. AD patients with BB-DNA above the median value (3.78 ng/mL) showed the lowest MMSE scores. No significant differences were observed in MCI subjects and EC. ANCOVA analysis adjusted for age, sex, education, and IADL was applied. * *p* < 0.05 as compared to BB-DNA_low_ in AD group.

**Figure 3 ijms-24-00078-f003:**
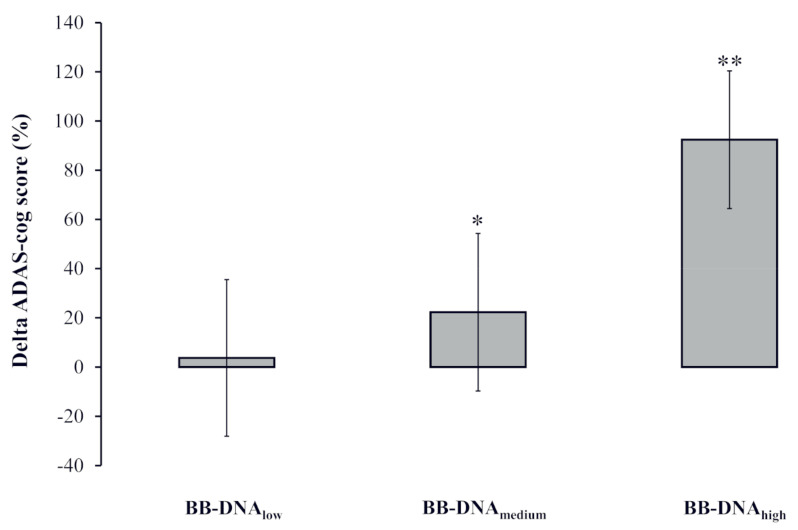
The change in the ADAS-cog score during the two-year follow-up in Alzheimer’s disease (AD) patients in accordance with blood bacterial DNA (BB-DNA) tertiles. Higher values of BB-DNA were associated with a worse cognitive decline after the two-year follow-up. ANCOVA analysis adjusted for age, sex, education, and IADL was applied. ** *p* < 0.001 as compared to BB-DNA_low_; * *p* < 0.05 as compared to BB-DNA_high_.

**Figure 4 ijms-24-00078-f004:**
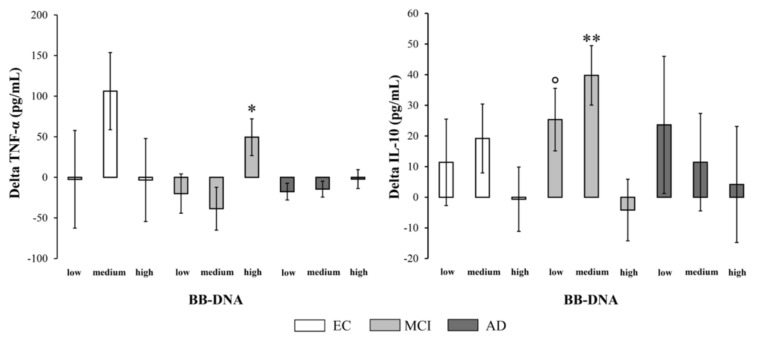
Changes in plasma TNF-α and IL-10 levels during the two-year follow-up in elderly controls (EC), mild cognitive impairment (MCI) and Alzheimer’s disease (AD) subjects in accordance with blood bacterial DNA (BB-DNA) tertiles. MCI patients showed a significantly higher TNF-α production and a significantly lower IL-10 production in BB-DNA_high_ than in the other tertiles (i.e., BB-DNA_medium_ and BB-DNA_low_) after a two-year follow-up. ANCOVA analysis adjusted for age, sex, education, and IADL was applied. ** *p* <0.01 as compared to BB-DNA_high_; * *p* < 0.05 as compared to BB-DNA_low_ and BB-DNA_medium_; ° *p* < 0.05 as compared to BB-DNA_high_.

**Table 1 ijms-24-00078-t001:** Characteristics of elderly controls (EC), mild cognitive impairment (MCI), and Alzheimer’s disease (AD) patients.

	EC *n* = 94	MCI *n* = 93	AD *n* = 95
**Age (years)**	72.7 ± 0.6	76.1 ± 0.6 *	77.8 ± 0.5 *
**Females**	79.6%	63.4% ^§^	67.7% ^§^
**BMI**	26.5 ± 0.5	25.3 ± 0.4	25.8 ± 0.4
**MMSE**	28.1 ± 0.1	25.9 ± 0.2 *^/^°	20.2 ± 0.3 *
**LSNS**	29.9 ± 0.9	29.2 ± 0.7	26.5 ± 0.7 **
**PASE**	110.6 ± 4.9 °	83.4 ± 4.5 °^/^*	64.6 ± 4.6
**Smoking habits**			
**never smoker**	58.5 %	62.3 %	56.8 %
**former smoker**	28.7 %	31.2 %	30.5 %
**current smoker**	12.8 %	6.5 %	12.7 %
**GDS**	7.6 ± 0.5	8.7 ± 0.5	8.3 ± 0.3
**ADL**	5.90 ± 0.02 °	5.90 ± 0.04 °	5.20 ± 0.10
**IADL**	7.90 ± 0.03 °	6.90 ± 0.10 °^/^*	3.30 ± 0.20
**Education (years)**	9.5 ± 0.5 °	5.9 ± 0.4 *	5.0 ± 0.3
**Albumin (g/dL)**	4.28 ± 0.03	4.23 ± 0.03	4.20 ± 0.03
**CRP (pg/mL)**	0.29 ± 0.07	0.34 ± 0.05	0.35 ± 0.05
**COPD**	3.2%	0.0%	5.3%
**Hypertension**	52.1%	53.8%	67.4%
**Diabetes**	4.3%	7.5%	7.4%
**CHD**	7.6%	8.6%	8.4%
**Atrial fibrillation**	18.1% °	11.1%	5.3%
**CKD**	1.1%	1.1%	3.2%
**Previous ictus**	1.1%	7.5%	10.5% ^§^
**PAD**	8.5%	8.6%	27.4% ^+^
**IBD**	3.2%	1.1%	0.0%
**Peptic ulcers**	2.1%	4.3%	0.0%
**Gastritis**	22.3%	7.5%	6.3%
**Acetylcholinesterase inhibitors**	0.0%	0.0%	42.5%
**Benzodiazepines**	24.7%	9.5% ^§^	12.5% ^§^
**Antidepressants**	2.6%	16.2% ^§^	18.8% ^§^
**Lipid-lowering medications**	14.3%	21.6%	18.8%
**Anticoagulants/antiplatelets drugs**	37.7%	51.4%	57.5% ^§^
**NSAIDs**	26.0%	27.0%	41.3%
**Antihypertensives**	51.9%	55.4%	57.5%
**Corticosteroids**	1.3%	0.0%	0.0%

BMI: body mass index; MMSE: mini mental state examination; LSNS: lubben social network scale; PASE: physical activity scale for elderly; GDS: geriatric depression scale; ADL: activities of daily living; IADL: instrumental activities of daily living; CRP: C-reactive protein; COPD: chronic obstructive pulmonary disease; CHD: coronary heart disease; CKD: chronic kidney disease; PAD: peripheral artery disease; IBD: inflammatory bowel disease; NSAIDs: nonsteroidal anti-inflammatory drugs; NS: not significant. ANCOVA analysis correcting for age and sex. * *p* < 0.0001 as compared to EC; ^§^ *p* < 0.05 as compared to EC; ° *p* < 0.01 as compared to AD; ** *p* < 0.05 as compared to EC and MCI; ^+^ *p* < 0.0001 as compared to EC ad MCI.

**Table 2 ijms-24-00078-t002:** Multivariate linear regression analysis for variables independently associated with blood bacterial DNA (BB-DNA) in Alzheimer’s disease patients stratified by Clinical Dementia Rating scale (CDR).

	Variables	Unstandardized Coefficients		Standardized Coefficients	*p* Value
		β	Std. Error	β	
**CDR1**	Sex	25.263	11.974	0.368	0.043
	Age	0.908	0.994	0.146	0.368
	AChEIs	14.495	10.931	0.219	0.195
	Benzodiazepines	−3.584	13.775	−0.041	0.796
	Antidepressants	26.822	17.021	0.264	0.125
	Lipid lowering medications	−3.537	13.218	−0.044	0.791
	Plasma BDNF levels	5.026	5.472	0.151	0.365
	Alcohol consumption	2.399	12.072	0.032	0.844
	Smoking_habits	−0.231	8.113	−0.005	0.977
	PASE	−0.092	0.142	−0.109	0.523
**CDR2**	Sex	17.329	15.439	0.231	0.272
	Age	0.343	1.218	0.054	0.780
	AChEIs	−8.667	12.175	−0.119	0.483
	Benzodiazepines	−21.675	23.614	−0.178	0.367
	Antidepressants	0.344	12.037	0.005	0.977
	Lipid lowering medications	−25.841	18.204	−0.266	0.168
	Plasma BDNF levels	8.375	3.430	0.407	**0.022**
	Alcohol consumption	−22.000	13.634	−0.284	0.119
	Smoking_habits	−13.065	10.939	−0.251	0.243
	PASE	−0.009	0.152	−0.010	0.954

AChEIs: Acetylcholinesterase inhibitors; PASE: physical activity scale for elderly. Sex was categorized as follows: 0 = males and 1 = females.

## Data Availability

The datasets analyzed in this study are available from the corresponding author upon reasonable request.

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
