# Peer review of "Bacterial DNAemia in Alzheimer’s Disease and Mild Cognitive Impairment: Association with Cognitive Decline, Plasma BDNF Levels, and Inflammatory Response"

_ijms, 2022, doi:10.3390/ijms24010078_

Round 1

Reviewer 1 Report

1. introduction and rationale of the study is not convincing 

2.  Inclusion and exclusion criteria is not defined properly in MM

3. The mean age for HC is 72.07+.0.6, so what does it mean healthy controls

4. MMSE in AD is 20.2±0.3* at the age of 77.8±0.5*, 

5. AD N= 95, authors took how many years to include such a large size, waht is the power of the sample size.

6. what is the purpose of including smokers in the study

7.satistical analysis is not robustic.

Author Response

  1. Introduction and rationale of the study is not convincing 

The introduction section was widely revised clarifying the rationale of the study.

  1. Inclusion and exclusion criteria is not defined properly in MM

We reported the inclusion and exclusion criteria in details in the Material and Method section (page 10): “The inclusion criteria were the following: age 65 or older, availability  during testing phases, the presence of a caregiver for subjects with cognitive decline (specific criterion for subjects with MCI and AD), and the ability to sign the informed consent (subjects who were not able to speak, were incapacitated or in need of a support administrator). The ex-clusion criteria were the following: serious medical problems (i.e., recent cardiovascular events, cancer, infections or acute renal failure) or major psychiatric disorders (i.e., major depression, schizophrenia, and other psychiatric illnesses that could limit participation in the study), sensorimotor deficits, severe AD, and the presence of neurodegenerative disorders other than AD (e.g.,Parkinson’s disease, progressive supranuclear palsy, corti-co-basal degeneration, Lewy’s body dementia, Huntington's disease, frontotemporal de-mentia, Pick's disease) cerebral hypoxia (acute or chronic), infections of the CNS (i.e., brain abscess, meningitis, AIDS), Creutzfeldt-Jacob disease, brain cancers, untreated epi-lepsy, and of alcohol or drug abuse in the past year.

  1. The mean age for HC is 72.07+.0.6, so what does it mean healthy controls

We apologize, but we did not understand the Reviewer's comment. Is he/she asking whether subjects with a mean age of 72 years are healthy? As specified in the Material and Methods section, healthy elderly controls do not have a diagnosis of cognitive decline, although, as shown in Table 1, may have other pathologies. To avoid any misunderstanding, we changed the abbreviation HC (healthy controls) to EC (elderly controls).

  1. MMSE in AD is 20.2±0.3* at the age of 77.8±0.5*

Since patients with severe AD are excluded from the study, the MMSE score in our patients with a mean age of 77.8 years falls in the 95% confidence interval (i.e., 19.2-20.9).

  1. AD N= 95, authors took how many years to include such a large size, what is the power of the sample size.

Study participants were enrolled between June 2012 and October 2014. We specified the enrolling period in the Material and Methods section (page 10).

In the comparison of BB-DNA among groups, the observed power was 0.73, η2 = 0.03. However, the post-hoc power analysis could be analytically misleading, and might not indicate true power for detecting statistical significance (PMID: 31552383). Thus, to avoid bias and highlight the issue rightly raised by the Reviewer, the following sentence has been added at the end of the Discussion section: ” Additional research on larger cohorts is also required to confirm our findings” (page 10)

  1. what is the purpose of including smokers in the study

In a previous study we found an increment of BB-DNA in smokers (PMID: 35914804). We added a new figure (Fig. 1S) showing BB-DNA level in EC, MCI, and AD subjects subdivided for smoking habits to clarify any possible correlation. Moreover, we added the following sentence in the Discussion section: “In contrast to our previous findings [Giacconi et al., 2022], no significant differences were observed in smokers from each group. This may be due to the smaller sample size analyzed or the different age range compared to the previous study, or the influence of other environmental factors.” (page 9)

Statistical analysis is not robustic.

We revised the statistical analysis and added the following part: “The differences in BB-DNA between groups were analyzed by ANCOVA analysis and generalized linear models (linear model with log-transformed values and identity link-function) after adjusting for age, sex, education, previous ictus, peripheral artery disease and drug use. Linear regression analysis using the stepwise method included the following covariates: sex, age, AChEI, benzodiazepines, antidepressants, lipid-lowering medications, alcohol consumption, smoking, PASE, BMI, education, IADL, GDS, and the ADAS-cog score. Each variable that met the criteria for removal was eliminated in a stepwise manner. Only variables below the removal threshold were retained in the final step.” (page 12)

The English language has been revised.

All the references have been checked and supplemented with new references in red.

Reviewer 2 Report

This study by Giacconi R. et al. measured BB-DNA in healthy elderly controls, mild cognitive impairment (MCI), and AD patients to explore the effect on plasma BDNF (pBDNF) levels and inflammatory response, and the association with cognitive decline during a two-year follow-up. In general, the study is novel and the results are interesting about microbial dysbiosis, bacterial translocation, systemic inflammation, BBB rupture, and neurodegeneration. However, my main concerns are related to the methodological aspects

Major concerns

1. In the exclusion criteria, the characteristics of patients with severe AD and serious medical conditions are unclear, please clarify. In addition, authors must specify the time of evolution of the patients with AD and MCI, and if only sporadic AD patients were included.

2. Since comorbidities such as diabetes, obesity, and metabolic syndrome, among others; gastrointestinal diseases such as ulcerative colitis, and Crohn's disease; and other chronic degenerative diseases (cardiovascular and renal) can cause microbial dysbiosis, it is necessary that the authors add to the table if the population studied in the three groups presents them and also perform the correlation. Besides, describe if the comorbidities of the study subjects were considered in their inclusion or exclusion criteria.

3. The authors must justify why they studied only IL-10, as an anti-inflammatory molecule, and TNF-α, as a pro-inflammatory molecule. Since differences were only found in the MCI group concerning IL1-0 and TNF-α and their association with circulating bacterial DNA (BB-DNA) tertiles, the authors need to discuss that other pro- and anti-inflammatory mediators could also have a pathological role in intestinal dysbiosis in patients with MCI and AD, as well as its possible association with the BBB dysfunction, peripheral immune infiltrates and glial activation (https://www.ncbi.nlm.nih.gov/pmc/ articles/PMC8914070/ https://pubmed.ncbi.nlm.nih.gov/33670754/).

4. Although the sub-section on “Plasma cytokine levels” describes the number of patients studied by gender, I suggest that they also mention it in section “4.1. Participants”. Given that the inflammatory phenotype can be influenced by gender, it would be important to differentiate between sexual dimorphism.

5. The authors describe that they analyzed their variables for 2 years (longitudinal), however, the periodicity of these measurements and their impact on the results are not clear.

Minor concerns

1. In figure 2 it is missing to indicate what the asterisk means

2. Lines 87-90 do not mention anticoagulants.

3. Check if references 16 and 17 are appropriate for that paragraph.

Author Response

This study by Giacconi R. et al. measured BB-DNA in healthy elderly controls, mild cognitive impairment (MCI), and AD patients to explore the effect on plasma BDNF (pBDNF) levels and inflammatory response, and the association with cognitive decline during a two-year follow-up. In general, the study is novel and the results are interesting about microbial dysbiosis, bacterial translocation, systemic inflammation, BBB rupture, and neurodegeneration. However, my main concerns are related to the methodological aspects

Major concerns

  1. In the exclusion criteria, the characteristics of patients with severe AD and serious medical conditions are unclear, please clarify. In addition, authors must specify the time of evolution of the patients with AD and MCI, and if only sporadic AD patients were included.
  2. We reported the inclusion and exclusion criteria in details in the Material and Method section (page 10): “The inclusion criteria were the following: age 65 or older, availability during testing phases, the presence of a caregiver for subjects with cognitive decline (specific criterion for subjects with MCI and AD), and the ability to sign the informed consent (subjects who were not able to speak, were incapacitated or in need of a support administrator). The ex-clusion criteria were the following: serious medical problems (i.e., recent cardiovascular events, cancer, infections or acute renal failure) or major psychiatric disorders (i.e., major depression, schizophrenia, and other psychiatric illnesses that could limit participation in the study), sensorimotor deficits, severe AD, and the presence of neurodegenerative disorders other than AD (e.g.,Parkinson’s disease, progressive supranuclear palsy, corti-co-basal degeneration, Lewy’s body dementia, Huntington's disease, frontotemporal de-mentia, Pick's disease) cerebral hypoxia (acute or chronic), infections of the CNS (i.e., brain abscess, meningitis, AIDS), Creutzfeldt-Jacob disease, brain cancers, untreated epi-lepsy, and of alcohol or drug abuse in the past year.
  3. Since comorbidities such as diabetes, obesity, and metabolic syndrome, among others; gastrointestinal diseases such as ulcerative colitis, and Crohn's disease; and other chronic degenerative diseases (cardiovascular and renal) can cause microbial dysbiosis, it is necessary that the authors add to the table if the population studied in the three groups presents them and also perform the correlation. Besides, describe if the comorbidities of the study subjects were considered in their inclusion or exclusion criteria.

We thank the Reviewer for the suggestion. We added in Table 1 the main comorbidities present in the studied population. Chronic inflammatory bowel diseases are not present in AD patients, while only 6.6 % of AD patients had gastritis.

  1. The authors must justify why they studied only IL-10, as an anti-inflammatory molecule, and TNF-α, as a pro-inflammatory molecule. Since differences were only found in the MCI group concerning IL1-0 and TNF-α and their association with circulating bacterial DNA (BB-DNA) tertiles, the authors need to discuss that other pro- and anti-inflammatory mediators could also have a pathological role in intestinal dysbiosis in patients with MCI and AD, as well as its possible association with the BBB dysfunction, peripheral immune infiltrates and glial activation (https://www.ncbi.nlm.nih.gov/pmc/ articles/PMC8914070/ https://pubmed.ncbi.nlm.nih.gov/33670754/).

We thank the Reviewer for the suggestion. We added the following sentences in the Discussion section (page 10):

“This study did not examine other pro- and anti-inflammatory mediators that may play a relevant role in intestinal dysbiosis and neurodegenerative diseases.  For instance, during chronic microbial dysbiosis certain bacterial species and their metabolites may trigger neuroinflammatory pathways promoting Aβ accumulation [Cattaneo et al., 2017, Padhi et al., 2022]. The interaction between various species of Aβ with receptors in glial cells induces the release of proinflammatory cytokines, leading to BBB dysfunction through increasing permeability, inducing structural changes in brain capillaries, and enhancing the migration of immune cells [Soto-Rojas et al., 2021]. Moreover, other anti-inflammatiory cytokines may play a protective role in dementia, such as IL-4, whose high circulating levels are associated with better cognitive function [Pagoni et al., 2022]. However, contrasting results exists in literature on the role of systemic inflammatory mediators as AD risk factors [Yeung et al., 2021] and the topic deserves further investigations.”

We also highlighted that (page 10): “Third, plasma cytokines were assessed in a subgroup of patients, and the sample size may have been too small; other pro- and anti-inflammatory mediators should also be assayed to strengthen the findings of TNF-α and IL-10”

  1. Although the sub-section on “Plasma cytokine levels” describes the number of patients studied by gender, I suggest that they also mention it in section “4.1. Participants”. Given that the inflammatory phenotype can be influenced by gender, it would be important to differentiate between sexual dimorphism.

As suggested by the Reviewer, we described the subgroups analyzed for the assay of the plasma cytokines in the “Participants” section, and, accordingly, we added Table 3Sb in the Supplementary Material.

Plasma cytokine levels at baseline and after the 2-year follow-up in elderly controls (EC), Mild Cognitive Impairment (MCI) and Alzheimer’s disease (AD) patients were stratified by sex. No significant differences were observed except for the increased baseline values of IL-10 in EC males as compared to MCI and AD males. These results were reported in the Result and Discussion sections (pages 4 and 9): “There was no significant difference in cytokine levels between the cohorts (Table 3Sa, Supplementary Material), either at baseline or after 2 years of follow-up. Only EC males exhibited significantly higher IL-10 values at baseline compared to MCI and AD males (p<0.05, Table 3Sb, Supplementary Material).”; “In this study, we measured plasma TNF-α and IL-10 levels at baseline and after a two-year follow-up period. In contrast to other research no significant differences were identified between EC, MCI, and AD subjects [35, 36], although baseline IL-10 values were higher in EC males compared to MCI and AD males.”

The authors describe that they analyzed their variables for 2 years (longitudinal), however, the periodicity of these measurements and their impact on the results are not clear.

We apologize, but we are not sure that we have correctly interpreted the Reviewer's concern. With regard to periodicity, cognitive assessments were carried out at baseline and after 2 years. We therefore exclude any carryover effect on cognitive tests. In the Material and Methods section (“Participants”), we better specified at which times the data were collected and analyzed: “Laboratory parameters, cytokine plasma levels, pBDNF, and a comprehensive clinical and neuropsychological assessment were carried out at baseline and after two years; whereas, BB-DNA was only determined at baseline.”

Minor concerns

  1. In figure 2 it is missing to indicate what the asterisk means

We reported in the legend to Figure 2 that “*, p < 0.05 as compared to BB-DNAlow in AD group.”

  1. Lines 87-90 do not mention anticoagulants.

We added the following sentence (page 3): “Data were adjusted for age, sex, education, previous ictus, peripheral artery disease and drug use.”

  1. Check if references 16 and 17 are appropriate for that paragraph.

We changed the references 16 and 17 (now numbered as 14 and 18 respectively) with Tauber et al., 2009 (PMID: 19918123) and Chen et al., 2020 (PMID: 32841647). All the references have been checked and supplemented with new references in red.

The English language has also been revised.

Round 2

Reviewer 2 Report

Good job done by the authors, I only suggest improving the quality of their figures.

Author Response

We  thank the Reviewer for his suggestions, we carefully edited the text and increased the quality of the figures changing the resolution to 600 dpi ( figures in tif format loaded as files separately and also embedded in the main manuscript).
